# Compositional and Morphological Characterization of ‘Sorrento’ and ‘Chandler’ Walnuts

**DOI:** 10.3390/foods11050761

**Published:** 2022-03-06

**Authors:** R. Romano, L. De Luca, M. Vanacore, A. Genovese, C. Cirillo, A. Aiello, R. Sacchi

**Affiliations:** Department of Agricultural Sciences, University of Napoli Federico II, Via Università, 100, 80055 Portici, Italy; raffaele.romano@unina.it (R.R.); marco.vanacor@gmail.com (M.V.); alessandro.genovese@unina.it (A.G.); chiara.cirillo@unina.it (C.C.); alessandra.aiello@unina.it (A.A.); sacchi@unina.it (R.S.)

**Keywords:** *Juglans regia* L., fatty acids, total polyphenols, γ-tocopherol, healthy food

## Abstract

In Italy, most of the cultivated walnuts belong to the Sorrento ecotype, and they are considered commercially valuable due to their specific organoleptic characteristics. The aim of this study is to evaluate and compare the morphological and compositional characteristics of walnuts sampled from ‘Sorrento’ trees cultivated in different locations in Campania and trees of both the ‘Chandler’ and ‘Sorrento’ varieties derived from the same location. The results demonstrated that ‘Sorrento’ and ‘Chandler’ walnuts have different biometric characteristics and a different fat content, with the highest fat content being found in the ‘Sorrento’ variety. Regarding the fatty acid (FA) composition, the content of monounsaturated and saturated fatty acids (MUFAs and SFAs) was highest in the ‘Sorrento’ variety (from 13 to 15% for MUFAs and from 11 to 13% for SFAs), while the polyunsaturated fatty acids (PUFAs) content was highest in the ‘Chandler’ variety (77%). The total phenolics content (TPC) was highest in the ‘Sorrento’ variety (from 910 to 1230 mg GAE/100 g), while no difference in γ-tocopherol content was found. Furthermore, the influence of walnut area cultivation was shown for fat content, FA composition and TPC. Therefore, both walnut varieties demonstrated good nutritional properties considering the PUFAs and γ-tocopherol content.

## 1. Introduction

The English (or Persian) walnut belongs to the order Juglandiflorae, family Juglandaceae and genus *Juglans*, and it grows in temperate climate areas. The English walnut tree production in 2019–2020 was 965,402 metric tons, second only to almond [1]. China accounts for more than 50% of the total walnut production, followed by the USA and the European Union. In Italy, representing twelfth place worldwide, the area harvested for walnuts with shells was 4670 ha in 2019, while the production was 10,800 tons [2], and the region traditionally most linked to walnut cultivation is Campania [3]. The first Italian area to produce walnut as well as the typical and still most widespread variety (even in Italy) is the Sorrento variety, which is the most famous among the Italian walnut ecotypes and is cultivated in the Sorrento Peninsula as well as in other areas of the Campania region [4]. In recent years, other varieties of *J.regia* have also been cultivated, such as the Californian Chandler, which has adapted to the Italian climate [5]. Recently, walnuts have garnered the “superfood” label because of their high contents of omega-3 fatty acids, phytochemicals, antioxidant polyphenols and fiber [6]. The COVID-19 pandemic (associated with SARS-CoV-2) has accelerated this increasing trend of healthy food consumption, further pushing the good growth prospects of dried fruit in general [1]. The countries with the highest per capita consumption of walnuts in the world are the USA (1.24 kg), Israel (1.2 kg), France (1.02 kg), Germany (0.62 kg), and Italy, Greece and Holland (all 3 with 0.47 kg). The per capita consumption of countries with high production, such as China (0.056 kg) and India (0.04 kg), is clearly lower [7].

The walnut kernel has a high energy density (approximately 630 kcal/100 g) and a high nutrient content, including 52.0–77.5% lipids; 11.0–25.0% protein; 5.0–24.0% carbohydrates; 1.3–2.5% mineral elements; microelements (Fe, Cu, Zn and I), macroelements (Ca and Mg), fibers, flavonoids and sterols, as well as vitamins: E (46.0 mg/100 g), C (30–55 mg/100 g), B1 (0.34–0.8 mg/100 g) and lower amounts of vitamins B2, A and PP [8,9,10,11,12,13,14,15,16]. Walnuts are also an important source of tocopherols, mono- and polyunsaturated fatty acids (MUFAs and PUFAs) and saturated fatty acids (SFAs), phospholipids and polyphenols. Linoleic acid is the most abundant fatty acid, followed by oleic acid and linoleic acid [9,17,18,19].

It has been demonstrated that regular walnut consumption has effects on human health associated with a reduction in cardiovascular and coronary heart diseases, diabetes [20] and neurological disorders [21], due to the n-3 and n-6 PUFAs kernel content [22,23,24].

The kernel contains phenolic compounds considered bioactive compounds, because they have health-beneficial properties, including anti-inflammatory, anti-mutagenic, anti-atherogenic and antioxidant effects [25]. Furthermore, some potential relationships between walnut consumption and treatments preventing different types of cancers have been investigated [26].

The kernel of walnut represents approximately 40–60% of the nut weight and contains approximately 52–72% oil depending on the variety [27,28], geographic location and type of irrigation and fertilization [29].

Thus, walnut characteristics depend on genotype, cultivation area and ecological and technological factors [30,31,32,33]. In Italy, walnuts are cultivated in several areas throughout the national territory; however, nuts produced in the Campania Region (southern Italy), belonging to the Sorrento ecotype, have a long tradition and are still considered commercially valuable due to their specific organoleptic characteristics, confirmed by genetic criteria established by Foroni et al. [4]. Historically, the name of this ancient variety comes from the original area of cultivation, where walnuts were cocultivated with olives and grapes on the hill terraces and intercropped with citrus in the lowlands. The nuts have a long connection with the area, and Sorrento walnuts were already being grown and appreciated by the Romans. Remains of fossilized walnuts and carbonized trees have been discovered in Herculaneum, and paintings of walnut trees have been found in the Pompei archaeological site. Currently, Sorrento walnut is listed among the typical and traditional products of the Campania Region (L. R n. 238, 2016), and it is the object of the SlowFood Presidia politics, wherein it is referred to as Sorrento Peninsula Walnut [34].

However, in the last decades, the cultivation of Sorrento variety has been restricted to traditional area and cultivation systems, whereas the new orchards have been planted diffusely by using Californian varieties. The main reason of this trend is related to the higher productivity of Californian genotypes, due to their lateral-fruiting habit compared to terminal-fruit bearing varieties, such as Sorrento [35].

The aim of this study is to evaluate and compare the morphological and compositional characteristics of walnuts sampled from (a) trees of “Sorrento” var. grown in different locations of the Campania region; (b) trees of both Chandler and Sorrento var. from the same location (Capua). The walnuts were characterized in terms of biometric characteristics, moisture content, lipid content, fatty acid composition, total polyphenols and γ-tocopherol content. Furthermore, few studies were present in the literature pertaining to the biometric characteristics of the Sorrento variety; thus, walnuts were also analyzed in terms of these characteristics.

## 2. Materials and Methods

### 2.1. Chemicals

Methanol, 99% *n*-hexane, and gallic acid monohydrate were purchased from Carlo Erba (France); 2-propanol was purchased from Sigma–Aldrich Co. (Milano, Italy); Folin-Ciocalteu reagent was purchased from Fisher Chemical (UK); and sodium carbonate was purchased from Fluka (Darmstadt, Germany). A standard of γ-tocopherol ≥96% was purchased from Sigma–Aldrich Co. (Milano, Italy) and Supelco TM 37 component FAME mix was obtained from Merck (Darmstadt, Germany).

### 2.2. Sampling

Walnut samples (‘Sorrento’ variety and ‘Chandler’ variety) were collected in September-October 2020 from orchards located in different cultivation areas of the Campania Region. Samples of the ‘Sorrento’ variety were harvested in Somma Vesuviana (Naples, Italy), Capua (Caserta, Italy), Vico Equense (Naples, Italy), Massa Lubrense (Naples, Italy), Maddaloni (Caserta, Italy) and Cicciano (Naples, Italy), while samples of the ‘Chandler’ variety were harvested only in Capua (Caserta, Italy). After collection, damaged fruits were discarded. Table 1 presents the code of the samples shown.

The sampled walnuts were manually husked and air-dried to reach at least 12% R.H. at least and stored in their shells at room temperature for 1 month before the measurements and analyses.

### 2.3. Nut Biometric Measurements

A subsample of 30 nuts per main sample was subjected to biometric measurements of walnut size and shape using UPOV walnut descriptors [36] by determining the longitudinal diameter, the two perpendicular maximum equatorial diameters and the shell thickness at two corresponding positions of the shell with a micrometric caliper. Furthermore, the mean equatorial diameter (mean of measurement of two equatorial diameters), symmetry index (ratio between two equatorial diameters) and shape index (ratio between mean equatorial diameter and longitudinal diameter) were calculated.

### 2.4. Determination of Moisture Content

The moisture content of the samples was determined following the method of Poggetti et al. [37], with modification. Briefly, each sample was dried using an oven at 105 ± 2 °C until a steady weight was obtained. The moisture content was expressed as grams of water instead of total weight in grams (g/100 g).

### 2.5. Lipid Extraction

Lipid extraction was performed following the method of Beyhan et al. [12]. Walnut kernels (5 g) were ground into a powder using a knife mill (Grindomix M200, Retsch, Italia, Verdere Scientific Srl, Bergamo, Italy). Subsequently, the lipid was extracted using a Soxhlet extractor with *n*-hexane as the solvent. After 3.5 h of extraction, the obtained sample was weighed, and the lipid yield was determined.

The oils were placed in vials and stored at −20 °C until analysis.

### 2.6. Fatty Acid Composition

The determination of fatty acid composition was performed by analyzing fatty acid methyl esters (FAMEs) obtained after *trans*-esterification. Briefly, a 1% solution of lipid in hexane was prepared, and 300 µL of 2 M KOH in methanol was added per 1 mL of solution [38]. A total of 1 μL of the upper layer, containing the FAMEs, was injected into an Agilent Technologies 6890N gas chromatograph equipped with a capillary column (100 m × 0.25 mm inner diameter, film thickness of 0.20 μm) with a polystationary phase (90% biscyanopropyl/10% cyanopropylphenyl siloxane) (Supelco, Bellefonte, PA, USA), hydrogen flame ionization detector (FID) and programmed temperature vaporizer (PTV). The temperature program was carried out as follows: the initial oven temperature was 140 °C for 5 min, which was increased by 4 °C min^−1^ to 175 °C and held for 20 min and then increased to 240 °C at 3 °C min^−1^ and maintained for 20 min. The PTV temperature was 60 °C for 0.1 min, and the PTV temperature was raised at 500 °C min^−1^ to 260 °C and held for 5 min. The FID detector and the injector temperatures were fixed at 260 °C and 120 °C, respectively, and the split ratio was 15:1. Helium was used as the carrier gas at 2 mL min^−1^ flow, and chromatographic air and hydrogen were used as auxiliary gases.

The chromatogram peaks were identified using an external 37-component standard (TM 37 component FAME mix, Supelco, Bellefonte, PA, USA) by comparing the retention times of the standards with those of the samples under the same operating conditions. The results were expressed as % *w*/*w*.

### 2.7. Determination of γ-tocopherol

The content of γ-tocopherol was quantified following the method proposed by Grilo et al. [39], with modifications. Lipid was weighed (50 mg), dissolved in 2 mL of isopropanol, shaken for 1 min with a vortex, filtered through a 0.45 µm PTFE filter and and then determined using an HPLC method. The system, an HPLC Agilent 1100 series, with a quaternary pump, a G4225A degaser, a G1315B diode array detector (DAD) and a Spherisorb ODS2 reversed-phase column, particle size 5 µm, 4.6 mm × 250 mm was used. The mobile phase was 100% methanol under isocratic conditions with a flow rate of 1 mL/min. The detector wavelength was set at 298 nm. The injection volume was 20 µL.

To quantify the concentration of the compound, calibration curves of the standard were constructed. The range of linearity was 10–200 mg/L, while the coefficient of determination (R^2^) was 0.998. The limit of detection (LOD) and limit of quantification were 10 and 20 mg/L, respectively. The results were expressed as mg/100 g of kernel.

### 2.8. Determination of Total Phenolic Content (TPC)

The extraction of phenolic compounds was determined following the method proposed by Slatnar et al. [40], with modifications. Briefly, 0.5 g of walnut was added to 10 mL of methanol:water (6:4, *v*/*v*), vortexed for 2 min and centrifuged at 8200× *g* for 10 min. The obtained supernatant was removed and filtered with a paper filter. The extraction procedure was repeated five times.

The total phenolic content (TPC) values of walnut extracts were determined using the Folin-Ciocalteu method, according to Labuckas et al. [41]. The reaction mixture contained 100 µL of polyphenol extract, 500 µL of Folin-Ciocalteu reagent and 2.5 mL of sodium carbonate (20% *w*/*v*). The final volume was made up to 10 mL with deionized water. After 1 h of reaction in the dark and at room temperature, the absorbance was measured at 725 nm with a Shimadzu UV 1601 spectrophotometer (Milan, Italy). The TPC was calculated with a calibration curve of gallic acid (50, 100, 200, 400 and 500 mg/L; R^2^ = 0.9996), and the results were expressed as milligrams of gallic acid equivalents (GAE)/100 grams of kernel.

### 2.9. Statistical Analysis

All analytical measurements were performed in three replicates, and the results were expressed as the mean values (± standard deviations).

Statistical analysis of data was performed by applying one-way analysis of variance (ANOVA), applying post hoc Tukey’s multiple-range test (*p* ≤ 0.05) and the principal component analysis (PCA). Data elaboration was carried out using XLStat (Version 2014.5.03), an add-in software package for Microsoft Excel (Addinsoft Corp., Paris, France).

## 3. Results and Discussion

### 3.1. Walnut Size and Shape

The nut biometric and weight measurements are shown in Table 2. Differences among the samples were found in all the parameters, except for the symmetry index. The longitudinal diameter of the Sorrento variety sampled from the orchards located in the different cultivation areas was highest in SML (40.22 mm), while the lowest value was registered in SCa (38.42 mm). Regarding the comparison of walnuts belonging to different *genotypes* but grown in the same location, the value for CCa was higher than that for SCa (46.11 mm vs. 38.42 mm).

The highest equatorial diameter was found in SSV (30.90 mm), while the lowest was found in SVE (29.28 mm). When comparing Sorrento and Chandler *var.* with respect to nut length, the equatorial diameter was higher in CCa than in SCa (37.75 mm vs. 29.94 mm).

The shape index was highest in SCi, SM and SCa (0.78), while the lowest values were observed in SVE and SML (0.75). Finally, intervarietal differences were found for ‘Chandler’ and ‘Sorrento’, with a higher value in CCa than in SCa (0.82 vs. 0.78).

Thus, the walnuts of the Chandler variety presented the highest sphericity compared to those of the Sorrento variety.

The nut biometric characterization is shown in Table 3. The value of shell thickness was highest in SCi (2.95 mm) and lowest in SCa (2.77 mm); no significant differences were found. Furthermore, when comparing the Chandler and Sorrento varieties, a higher value was found in Chandler (3.25 vs. 2.77 mm). All the obtained values were higher than those of walnut varieties (Xiangling and Jizhaomian) grown in China, which showed shell thickness values of 1.03 mm and 1.39 mm, respectively [42], and they were also higher than those of wild accessions of walnut cultivated in the Friuli Venezia Giulia region, which showed a value of 1.3 mm [37]. The shell thickness affects the ease of shell removal and, from a commercial point of view, is better for the quality of the nut [43]. Among the Sorrento varieties, the shell weight was highest in Sci (7.27 g) and lowest in SML (6.00 g); furthermore, an intervarietal difference was found, with a higher value for CCa than for SCa (8.57 vs. 6.25 g).

Regarding the total nut weight and the kernel weight, the value was highest in SCi (14.61 and 7.34 g, respectively) and lowest in SML (10.90 and 4.90 g, respectively). Intervarietal differences were also noticed, as the Chandler walnuts showed a higher value of both total nut weight and kernel weight than Sorrento walnuts (15.36 vs. 11.95 g and 6.79 vs. 5.70 g, respectively).

Finally, the percentage of kernel among total weight also ranged between 49.99% in SCi to 55.92% in SCa.

### 3.2. Moisture and Lipid Content

The moisture and lipid contents of walnuts are shown in Table 4.

The moisture content ranged from 3.2% in SML to 5.0% in SVE. These values were similar to Amaral et al. [28], who found a moisture content that ranged from 3.6% to 4.4% depending on the analyzed variety, while Pereira et al. [44] showed a moisture content that ranged from 3.9% to 4.5% depending on the analyzed variety. No significant difference between ‘Chandler’ and ‘Sorrento’ was found.

Regarding the lipid content, intravarietal and intervarietal differences were found. The lipid content ranged from 61% in SML to 66% in SCi, while regarding the walnuts cultivated in the same zone, the lipid content was higher in SCa than in CCa (66 vs. 62%). All the obtained values were lower than those reported by Martínez et al. [45], who reported a range from 70% to 73% in commercial Chandler varieties of different origins, while they ranged from 72% to 74% in Sorrento variety walnuts of different origins; Pereira et al. [44] showed a range from 69% to 72% in walnuts of different varieties. Furthermore, Liu et al. [42] showed lipid contents of 66% and 67% in the Xiangling and Jizhaomian varieties, respectively. The lipid content was in the range reported by Kafkas et al. [19] who reported a minimum value of 54% and a maximum value of 66% in different walnut varieties and genotypes grown in the USA.

Martínez and Maestri [9] showed a lipid content of 73% in the Chandler variety and 74% in the Sorrento variety. Finally, the value reported was similar to that found by Bada et al. [46], who reported an oil content that ranged from 55% to 68% depending on walnut origin.

The lipid content is influenced by genotype, but may also be influenced by environmental factors and irrigation management [44,45,47].

### 3.3. Fatty Acid Profile

The fatty acid composition of walnut is reported in Table 5.

Linoleic acid (C18:2) was present at the highest concentration, ranging from 58% in SVE to 61% in CCa, followed by oleic acid (C18:1), ranging from 13% in SCI to 15% in SSV, and linolenic acid (C18:3), ranging from 13% SM to 15% in SVE. These results were in accordance with Amaral et al. [28], who found a concentration of linoleic acid in the range of 58% to 63%, a range of oleic acid from 14% to 18% and linolenic acid ranging from 10% to 13% depending on the analyzed variety. Furthermore, Pereira et al. [44] reported concentrations that ranged from 56% to 60%, from 16% to 20% and from 13% to 17% for linoleic, oleic and linolenic acids, respectively. Ojeda-Amador et al. [48] found a concentration of linoleic acid that ranged from 60% to 62%, oleic acid from 15% to 16% and linolenic acid from 13% to 15%. In both studies, concentration was strongly influenced by the variety.

PUFAs were the most abundant fatty acids in all samples of walnuts analyzed, with a range from 73% in WSVE to 77% in CCa, while SFAs were the least concentrated, ranging from 10% in CCa to 13% in SVE, and the most abundant fatty acids were palmitic (C16:0) and stearic acid (C18:0), as reported by Liu et al. [42] and Tapia et al. [49]. The high content of PUFAs suggests susceptibility of oil to lipid oxidation reactions, thus reducing the shelf life [40]. The MUFA content ranged between 13% in CCi and 15% in SSV, similar to the results of Amaral et al. [28], Pereira et al. [44] and Ojeda-Amador et al. [48].

It was also calculated that the ratio between PUFA/SFA ranged from six in SVE and SM to eight in CCa. It is reported that PUFA/SFA is an index used to verify the impact of diet on cardiovascular health (CVH), and it is used to evaluate the nutritional value of dietary foods. In fact, all PUFAs in the diet can reduce low-density lipoprotein cholesterol (LDL-C) and reduce levels of serum cholesterol, while SFAs increase the levels of serum cholesterol. Therefore, a higher ratio has a positive effect on health [50].

The ratio between ω-6 and ω-3 ranged from four in SVE to five in CCa, in good agreement with the reports by other authors [9,28,51,52]. The ratio between ω-6 and ω-3 was favorable due to the high content of linolenic acid [49].

A study by Torabian et al. [53] demonstrated that the inclusion of walnuts in a diet positively influenced the plasma lipid profile, and it was shown that the inclusion of ω-3 and ω-6 PUFAs through the consumption of walnuts lowered cardiovascular disease risk [54].

Furthermore, the C18:1/C18:2 ratio (Table 5) was similar to that observed by Bouabdallah et al. [55], who also showed a possible relationship with the stability of oil; in fact, a high ratio was correlated with a long shelf life.

The results for the different varieties grown in the same location revealed a variation between the two varieties in terms of fatty acid composition; in particular, the Sorrento variety showed the highest contents of palmitic, oleic, and linolenic acids, while the Chandler variety showed the highest content of linoleic acid. Thus, MUFA and SFA were highest in the Sorrento variety, while PUFA was highest in the Chandler variety.

Differences in fatty acid composition could be explained by the different geographical origins or other factors, such as agricultural practices [9].

Furthermore, statistically significant differences among varieties were found with respect to the fatty acid profile (Table 5), as reported by Martínez and Maestri [9].

In particular, SCa showed higher contents of palmitic (C16:0) and oleic (C18:1) than CCa (8% vs. 7%; 14% vs. 13%, respectively), while CCa showed the highest content of linoleic acid (C18:2) (63% in CCa vs. 60% in SCa). MUFAs and SFAs were highest in SCa, while PUFAs were highest in CCa.

Therefore, considering the fatty acid composition and PUFA/SFA and (C18:1/C18:2) ratios, it is possible to conclude that the two walnut varieties exhibit excellent nutritional properties.

### 3.4. Total Polyphenol Content (TPC)

The TPC value ranged between 847 mg GAE/100 g in CCa and 1230 mg GAE/100 g in SSV. Intravarietal differences were found, while no intervarietal differences were found (Table 6).

The value was lower compared to Wu et al. [56] who reported a TPC of 1556 mg GAE/100 g, and Tapia et al. [49] who reported a TPC ranging from 2800 mg GAE/100 g to 5800 mg GAE/100 g depending on the analyzed variety, with a content of 5100 mg GAE/100 g in the Chandler variety. Furthermore, Christipoulos and Tsantili [57] reported a concentration of 2200 mg and GAE/100 g dry weight in the Chandler variety.

The measured value was also lower compared with Kornsteiner et al. [58], who found a TPC of 1625 mg GAE/100 g, while it was similar to the results reported by Slatnar et al. [40], including a range from 610 to 1093 mg GAE/100 g in kernels of different varieties, and similar to Pycia et al. [59], who reported TPC values that ranged from 820 mg GAE/100 g of dry weight to 2090 mg GAE/100 g of dry weight in various walnut varieties. For the Chandler variety, the results were similar to Fuentealba et al. [60], who reported TPC ranging from 570 mg/100 g dry weight to 1270 mg/100 g dry weight depending upon the color of the shelled walnuts. The TPC was influenced by the variety analyzed, as demonstrated by Kafkas et al. [19]. Furthermore, the differences between the TPC of the walnut samples may have been due to climatic variations, storage conditions, the harvesting period, or variety, as well as the differences between the varieties [61], and may be dependent on the solvent used for extraction [41]. Finally, pre- and postharvest factors significantly affected the TPC of walnuts [62].

### 3.5. γ-Tocopherol Content

In Table 6, the content of γ-tocopherol is shown. No intravarietal or intervarietal differences were found. The values ranged from 47.5 mg/100 g in SM to 63.9 mg/100 g in SML.

The content of γ-tocopherol was higher than the content reported by Lavedrine et al. [63], who reported concentrations ranging from 21.8 to 26.5 mg/100 g in the Franquette and Hartley varieties grown in different locations. Delgado-Zamarreno and coworkers [64,65] did not achieve separation between *β*- and γ-tocopherols by using RP-HPLC, so they quantified these two compounds together. Regardless, the values reported by Delgado-Zamarreno et al. [64,65] were lower than the γ-tocopherol contents detected in our samples (12.9 mg/100 g and 19.5 mg/100 g, respectively).

Additionally, Amaral et al. [17] found that γ-tocopherol content was affected both by the genotype and by the geographical origins of the analyzed varieties, ranging from 17.3 to 26.2 mg/100 g, while Beyhan et al. [12] reported γ-tocopherol concentrations in a range from 16.1 mg/100 g to 29.3 mg/100 g in several walnut genotypes grown in Turkey. However, the content of γ-tocopherol in walnuts can also be influenced by the year of production and by environmental factors [17]. The tocopherols inhibit lipid oxidation in vegetable oils and fats. The antioxidant activity of tocopherols is derived by donation of its phenolic hydrogen to both free radicals and decelerates the lipid peroxidation process [66]. In addition to its role as an antioxidant, Li et al. [67] reported that γ-tocopherol is effective in reducing platelet aggregation and LDL oxidation and delaying intraarterial thrombus formation.

### 3.6. PCA

Principal component analysis (PCA) was performed on all the analytical data to examine differences between walnut samples (Figure 1). The walnut varieties were clearly distinguished by PCA, and 76.10% of the total variance was explained by two first principal components, thus indicating that each walnut variety possesses some biometric and chemical peculiarities. The lipid content was different between the walnut samples, showing intra- and intervarietal differences, with the highest content found in ‘Sorrento’ variety. The PC1, in particular, shows the difference between two varieties, the PC2 among samples of ‘Sorrento’ walnuts in relation to the growing area. The composition of ‘Sorrento’ walnut, with respect to ‘Chandler’ ones shows: higher content of MUFAs, SFA and linolenic acid, lower content of linoleic acid and total PUFAs and higher level of antioxidants (TPC and tocopherols).

The differences among the ‘Sorrento’ samples suggested that site altitude has significant influence both on the biometric characteristics and on the composition of walnuts. Indeed, the SSV sample, showing the highest kernel percentage and the highest content in TPC were harvested in a site located at the highest altitude (Table 1), thus confirming the results of previous study on the positive relationships between polyphenolic content and altitude of the growing site in walnuts [68,69]. Furthermore, in walnuts a positive correlation between altitude and MUFA was also recorded, supporting the highest MUFA content detected in SSV and SML samples [70].

In conclusion, this study demonstrated that the ‘Sorrento’ and ‘Chandler’ walnuts have different lipid contents, with the highest content found in the Sorrento variety, and different fatty acids composition. The TPC showed intervarietal differences, while no difference for γ-tocopherol content was found.

The differences among the ‘Sorrento’ samples hinted that different altitude of the locations could influence both biometric characteristics and walnut composition.

However, both types of walnuts are a natural and readily available source of both natural antioxidants and PUFAs: they represent a food with good nutritional properties and they have an important role in the Mediterranean diet [71].

## Figures and Tables

**Figure 1 foods-11-00761-f001:**
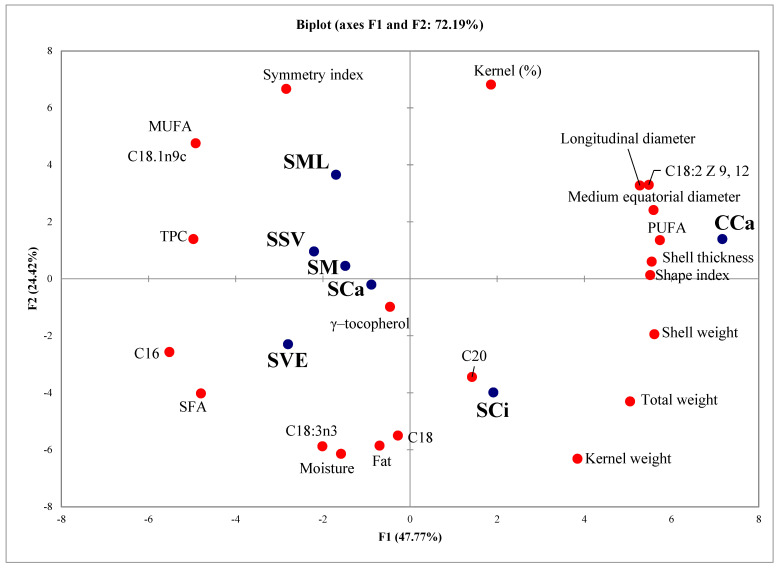
Results of principal component analysis (PCA) on Sorrento walnuts grown in different locations of Campania and Chandler walnut.

**Table 1 foods-11-00761-t001:** Code of samples used for analysis.

Variety	Location	Altitude (m.a.s.l.) *	Code
Sorrento	Cicciano (Na)	50	SCi
Sorrento	Vico Equense (Na)	90	SVE
Sorrento	Somma Vesuviana (Na)	165	SSV
Sorrento	Massa Lubrense (Na)	121	SML
Sorrento	Maddaloni (Ce)	73	SM
Sorrento	Capua (Ce)	25	SCa
Chandler	Capua (Ce)	25	CCa

* Meters above sea level.

**Table 2 foods-11-00761-t002:** Biometrical characteristics of walnuts (*n* = 30).

Code	Longitudinal Diameter (mm)	Medium Equatorial Diameter (mm)	Symmetry Index	Shape Index
SCi	39.14 ± 0.52 ^bc^	30.31 ± 0.36 ^bc^	0.927 ± 0.006 ^a^	0.78 ± 0.01 ^b^
SVE	39.01 ± 0.36 ^bc^	29.28 ± 0.22 ^c^	0.940 ± 0.005 ^a^	0.75 ± 0.01 ^c^
SSV	39.98 ± 0.33 ^b^	30.90 ± 0.26 ^b^	0.953 ± 0.008 ^a^	0.77 ± 0.01 ^b^
SML	40.22 ± 0.43 ^b^	29.93 ± 0.21 ^bc^	0.948 ± 0.007 ^a^	0.75 ± 0.01 ^c^
SM	39.22 ± 0.34 ^bc^	30.73 ± 0.19 ^b^	0.946 ± 0.004 ^a^	0.78 ± 0.01 ^b^
SCa	38.42 ± 0.34 ^c^	29.94 ± 0.24 ^bc^	0.944 ± 0.006 ^a^	0.78 ± 0.01 ^b^
CCa	46.11 ± 0.70 ^a^	37.75 ± 0.69 ^a^	0.940 ± 0.006 ^a^	0.82 ± 0.01 ^a^

Different letters on the same columns indicate statistically significant differences (*p* < 0.05).

**Table 3 foods-11-00761-t003:** Biometric characterization of walnuts (*n* = 30).

Code	Shell Thickness (mm)	Shell Weight (g)	Kernel Weight (g)	Total Weight (g)	Kernel (%)
SCi	2.95 ± 0.10 ^b^	7.27 ± 0.23 ^b^	7.34 ± 0.28 ^a^	14.61 ± 0.48 ^a^	49.99 ± 0.65 ^c^
SVE	2.81 ± 0.10 ^b^	6.87 ± 0.14 ^bc^	6.22 ± 0.26 ^bc^	13.09 ± 0.33 ^b^	53.15 ± 1.40 ^ab^
SSV	2.64 ± 0.10 ^b^	6.39 ± 0.11 ^cd^	5.92 ± 0.15 ^cd^	12.30 ± 0.24 ^bc^	52.06 ± 0.60 ^bc^
SML	2.88 ± 0.09 ^b^	6.00 ± 0.11 ^d^	4.90 ± 0.19 ^e^	10.90 ± 0.23 ^d^	55.45 ± 1.08 ^a^
SM	2.90 ± 0.10 ^b^	6.31 ± 0.21 ^d^	5.44 ± 0.14 ^de^	11.75 ± 0.25 ^cd^	53.25 ± 1.48 ^ab^
SCa	2.77 ± 0.11 ^b^	6.25 ± 0.13 ^d^	5.70 ± 0.16 ^cd^	11.95 ± 0.26 ^c^	52.44 ± 0.57 ^bc^
CCa	3.25 ± 0.12 ^a^	8.57 ± 0.27 ^a^	6.79 ± 0.26 ^ab^	15.36 ± 0.49 ^a^	55.92 ± 0.69 ^a^

Different letters in the same column indicate statistically significant differences (*p* < 0.05).

**Table 4 foods-11-00761-t004:** Kernel moisture and lipid content (*n* = 3).

Code	Moisture Content (% *w*/*w*)	Lipid Content (% *w*/*w*)
SCi	4.9 ± 0.12 ^a^	66 ± 0.33 ^a^
SVE	5.0 ± 0.06 ^a^	64 ± 0.21 ^b^
SSV	4.9 ± 0.06 ^a^	63 ± 0.22 ^bc^
SML	3.2 ± 0.06 ^d^	61 ± 0.13 ^d^
SM	3.5 ± 0.05 ^b^	66 ± 0.30 ^a^
SCa	3.4 ± 0.02 ^c^	66 ± 0.34 ^a^
CCa	3.5 ± 0.02 ^b^	62 ± 0.32 ^c^

Different letters in the same column indicate statistically significant differences (*p* < 0.05).

**Table 5 foods-11-00761-t005:** Fatty acid profile of kernel (*n*
*=* 3).

Fatty Acid	Sorrento						Chandler
	SSV (%)	SVE (%)	SML (%)	SM (%)	Sci (%)	SCa (%)	CCa (%)
C16:0	9 ± 0.04 ^bc^	10 ± 0.05 ^a^	9 ± 0.15 ^bc^	9 ± 0.30 ^b^	8 ± 0.08 ^c^	8 ± 0.09 ^c^	7 ± 0.01 ^d^
C18:0	2 ± 0.01 ^d^	3 ± 0.04 ^a^	2 ± 0.02 ^d^	3 ± 0.06 ^ab^	3 ± 0.02 ^c^	3 ± 0.03 ^b^	3 ± 0.02 ^c^
C18:1	15 ± 0.01 ^a^	14 ± 0.09 ^c^	14 ± 0.04 ^ab^	14 ± 0.02 ^ab^	13 ± 0.16 ^d^	14 ±0.02 ^bc^	13 ± 0.01 ^d^
C18:2	60 ± 0.10 ^c^	58 ± 0.07 ^d^	61 ± 0.17 ^b^	60 ± 0.07 ^c^	61 ± 0.02 ^b^	60 ± 0.13 ^c^	63 ± 0.01 ^a^
C20:1	0.30 ± 0.17 ^a^	0.24 ± 0.01 ^a^	0.20 ± 0.01 ^a^	0.3 ± 0.09 ^a^	0.3 ± 0.12 ^a^	0.2 ± 0.02 ^a^	0.3 ± 0.02 ^a^
C18:3	14 ± 0.12 ^bc^	15 ± 0.11 ^a^	14 ± 0.19 ^cd^	13 ± 0.20 ^d^	14 ± 0.01 ^ab^	14 ± 0.03 ^b^	14 ± 0.02 ^d^
MUFA	15 ± 0.01 ^a^	14 ± 0.09 ^c^	14 ± 0.04 ^ab^	14 ± 0.02 ^ab^	13 ± 0.16 ^d^	14 ± 0.02 ^bc^	13 ± 0.01 ^d^
PUFA	74 ± 0.22 ^c^	73 ± 0.18 ^d^	75 ± 0.02 ^c^	74 ± 0.13 ^d^	75 ± 0.02 ^b^	75 ± 0.16 ^c^	77 ± 0.01 ^a^
SFA	11 ± 0.22 ^c^	13 ± 0.09 ^a^	11 ± 0.02 ^c^	12 ± 0.15 ^b^	11 ± 0.18 ^c^	11 ± 0.14 ^c^	10 ± 0.01 ^d^
n-6/n-3	4 ± 0.03 ^b^	4 ± 0.02 ^c^	4 ± 0.07 ^a^	4 ± 0.07 ^a^	4 ± 0.01 ^b^	4 ± 0.01 ^b^	5 ± 0.01 ^a^
MUFA/PUFA	0.2 ± 0.01 ^a^	0.2 ± 0.01 ^a^	0.2 ± 0.01 ^a^	0.2 ± 0.01 ^a^	0.2 ± 0.01 ^a^	0.2 ± 0.01 ^a^	0.2 ± 0.01 ^a^
MUFA/SFA	1 ± 0.03 ^b^	1. ± 0.01 ^e^	1 ± 0.01 ^b^	1 ± 0.02 ^d^	1 ± 0.03 ^d^	1 ± 0.01 ^c^	1 ± 0.01 ^a^
PUFA/SFA	7 ± 0.15 ^c^	6 ± 0.10 ^e^	7 ± 0.01 ^b^	6 ± 0.10 ^d^	7 ± 0.11 ^c^	7 ± 0.10 ^c^	8 ± 0.01 ^a^
C18:1/C18:2	0.2 ± 0.01 ^a^	0.2 ± 0.01 ^a^	0.2 ± 0.01 ^a^	0.2 ± 0.01 ^a^	0.2 ± 0.01 ^b^	0.2 ± 0.01 ^a^	0.21 ± 0.01 ^b^

Different letters in the same row indicate statistically significant differences (*p* < 0.05).

**Table 6 foods-11-00761-t006:** Total phenol content (TPC) and γ-tocopherol content of kernels (*n* = 3).

Code	TPC mg GAE/100 g of Kernel	γ-Tocopherol mg/100 g of Kernel
SCi	970 ± 6.09 ^e^	59.5 ± 6.37 ^a^
SVE	1153 ± 2.11 ^b^	60.9 ± 7.26 ^a^
SSV	1230 ± 1.07 ^a^	48.5 ± 0.89 ^a^
SML	1138 ± 2.46 ^c^	63.9 ± 6.57 ^a^
SM	1102 ± 0.72 ^d^	47.5 ± 3.71 ^a^
SCa	910 ± 2.50 ^f^	62.2 ± 2.18 ^a^
CCa	847 ± 4.21 ^f^	54.1 ± 6.05 ^a^

Different letters in the same column indicate statistically significant differences (*p* < 0.05).

## Data Availability

Additional material is available upon request.

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
