# Peer review of "Compositional and Morphological Characterization of ‘Sorrento’ and ‘Chandler’ Walnuts"

_foods, 2022, doi:10.3390/foods11050761_

Round 1

Reviewer 1 Report

The manuscript is very well written, well designed, easy to follow and valuable both economically and especially medically - for the prevention of many modern diseases.  small remarks:

 - Please add some significant botanical characteristics that differentiate the two varieties: Sorrento variety and Chandler variety, in addition to those related to fruit (eg height, shape / number of leaflets, smell, etc.).

- Please add some comparative phytochemical data between the two studied varieties.

-Conclusions could be improved.

Author Response

Hi,

thank you for your suggestions. In Attachment you will find our answers. In the manuscript you find all the changes made.

Sincerely,
Lucia De Luca

Reviewer 2 Report

It is opinion of the reviewer needs several corrections/modifications. My individual comments are listed below.

Capitals letters in the paper title.

5-6 - Authors’ e-mail addresses and initials.

18 – It should be  “total phenolics content”.

19 – The results without any digitals after decimal point.

22 – What does it mean “good nutritional properties”?

26 – “Juglandiflorae” and “Juglandaceae” not with italic.

47 – I suggest to use throughout entire paper a term of “lipids” instead of “fat”.

48 – Microelements belong to mineral elements, Ca and Mg are macroelements.

53 – It should be [9,17-19].

57 – It should be [22-24].

63 – It should be [29-32].

Introduction. The biological activity of bioactive compounds of walnuts should be described.

117-122 – This is too trivial and must be omitted.

133 – It should be “2 M”.

167 – The centrifugation must be characterized by “x g” instead of “rpm”.

151 – It should be “γ-Tocopherol”.

153 – It should be “and then determined using an HPLC method”.   

159 – The authors should present typical HPLC chromatogram.  What about other tocopherols? In my opinion 100% methanol as a mobile phased did not offer good condition for tocopherols separation.

159 – The wavelength of emission?

161 – ppm is not any unit of SI.

162 – R2 is a determination coefficient.

167 – The centrifugation must be characterized “x g” instead of “rpm”.

Table 2,3,4 -  Information ” Different letters on the same columns indicate statistically significant differences (p<0.05)” must be moved to the table footnote.

Table 4 – For the moisture, the highest result must be marked with “a”, lower with “b”, etc.

309 – It should be “Phenolic compounds’.

310, 364, 365, Table 6 – TPC should be reported without any digitals after decimal point. γ-Tocopherol with one digital.

In my opinion the Conclusions should be completed.

385 – It should be  “References”.

References – The abbreviations instead of full titles. The Latin names must be written with italic.

428 – It should be   “J Agric Food Chem”.

Author Response

Hi,

thank you for your suggestions. In attachment you will find our answers. In the manuscript you found all the changes made.

Sincerely,

Lucia De Luca

Round 2

Reviewer 2 Report

The authors corrected this paper properly taken under considerations all my comments. Therefore, I can accept it now.

Author Response

Hi,

we apologize for the misunderstanding. We provided to solver all the  requests.Even though we must say the line numbers in our file do not match with the line number of the PDF file.We tried to solve the problem but we just couldn't.

Attached you will find our file with the replies to requests .New modification applied are blue-coloured.

Sincerely,

Lucia De Luca
